# Cerebrolysin Prevents Brain Injury in a Mouse Model of Liver Damage

**DOI:** 10.3390/brainsci11121622

**Published:** 2021-12-09

**Authors:** Shandiz Morega, Bogdan Cătălin, Cristiana Eugenia Simionescu, Konstantinos Sapalidis, Ion Rogoveanu

**Affiliations:** 1U.M.F. Doctoral School Craiova, University of Medicine and Pharmacy of Craiova, 200349 Craiova, Romania; morega.shandiz@yahoo.com; 2Experimental Research Centre for Normal and Pathological Aging, University of Medicine and Pharmacy of Craiova, 200349 Craiova, Romania; 3Department of Pathology, University of Medicine and Pharmacy of Craiova, 200349 Craiova, Romania; 43rd Department of Surgery, AHEPA University Hospital, Medical School, Aristotle University of Thessaloniki, 54124 Thessaloniki, Greece; sapalidis@med.auth.gr; 5Gastroenterology Department, University of Medicine and Pharmacy of Craiova, 200349 Craiova, Romania; ionirogoveanu@gmail.com

**Keywords:** liver transplantation, neurological complications, neuroprotection

## Abstract

Liver damage can lead to secondary organ damage by toxic substances and catabolic products accumulation which can increase the permeability of blood-brain barrier, leading to cognitive impairment. The only real treatment for end stage liver failure is grafting. With some, but not all, neurological symptoms subsiding after transplantation, the presence of brain damage can impair both the short and long-term outcome. We tested if Cerebrolysin can prevent brain injury in an experimental model of non-viral liver damage in mice. Behavior, abdominal ultrasound evaluation and immunohistochemistry were used to evaluate the animals. No ultrasound or behavior differences were found between the control and treated animals, with both groups displaying more anxiety and no short-term memory benefit compared to sham mice. Cerebrolysin treatment was able to maintain a normal level of cortical NeuN^+^ cells and induced an increase in the area occupied by BrdU^+^ cells. Surprisingly, no difference was observed when investigating Iba1^+^ cells. With neurological complications of end-stage liver disease impacting the rehabilitation of patients receiving liver grafts, a neuroprotective treatment of patients on the waiting lists might improve their rehabilitation outcome by ensuring a minimal cerebral damage.

## 1. Introduction

Liver damage is an important cause of mortality and morbidity worldwide. In 2016, cirrhosis was reported to be the 11th leading cause of death and 15th cause of morbidity [1]. Since the liver plays a key role in detoxification, liver dysfunction, regardless of the cause, leads to an accumulation of certain substances and catabolic products [2]. With this increase, a wide range of neurological and neuropsychiatric manifestations can occur [3,4]. Clinical and experimental data have shown that cortical changes are observed in all forms of liver damage, but it is still unclear which cellular and molecular signaling pathways are affected and at what extend. Since around approximately 30% of normo-ponderal and almost 80% of all overweight people are affected by non-alcoholic fatty liver disease/non-alcoholic steatohepatitis (NAFLD/NASH), this has developed into a silent epidemics in the last few decades [5,6,7]. With neurological and neuropsychiatric manifestations that vary from acute manifestations (encephalitis, myelitis, encephalomyelitis or Guillain-Barré syndrome [8,9]) to chronic manifestations such as cognitive impairment and dementia [10,11], the exact molecular pathways of this alterations are not completely understood. Theories explaining cortical changes are incriminating the altered permeability of blood-brain barrier (BBB) which allow a large number of small polar molecules to enter the brain, ultimately leading to edema [12,13,14,15]. However, insulin resistance, inflammation, hormonal alterations and change in levels of secreted hepatokines [16] have also been reported in this patients.

Some of them will ultimately need a liver transplantation (LT), since grafting is the only definitive treatment for end-stage liver disease, independent of the ethology [17,18]. After transplantation, the BBB permeability impairment can normalize. Regardless, the already acquired brain damage can be permanent. Since 10 to 85% of LT recipients experience central nervous system (CNS) complications (ranging from focal to diffuse brain injury), the main issue with this types of patients is that CNS damage can impair both the short and long-term outcome [19]. In addition, pretreatment with calcineurin inhibitors drugs may exacerbate dysarthria, akinetic mutism, confusion and seizures [20,21,22].

Given all evidence of cortical changes in patients with liver damage [5,23] and the long-term implication of such changes, a neuroprotective treatment may be considered. Clinical and experimental studies have shown that in other neurological condition like stroke [24], some encephalopathies [25], especially experimental autoimmune encephalomyelitis [26], certain neuroprotective strategies, such as using a Cerebrolysin, a cocktail of low molecular weight peptides and amino acids [27,28]. As the exact mechanisms of its neuroprotective and neurotrophic effects are heterogenous. Reports have attributed its neurotrophic effect to imitations, seemingly through activating the e PI3K/Akt pathway, in a similar manner to brain derived neurotrophic factor (BDNF) [29,30,31]. Cerebrolysin’s neuroprotective effects have been largely attributed to its role in activating the Sonic Hedgehog signaling pathway [32], but reports have also shown that it is able to lower free radical levels in the cortex and reduce the amount of proapoptotic enzymes [33]. With multiple mechanism involved in the cerebral alterations of liver damage, a single molecular pathway approach for neuroprotection seems not appear to be feasible, but with multiple ways in which Cerebrolysin increases neuroprotection, it seems like a perfect candidate for the task be.

## 2. Materials and Methods

### 2.1. Experimental Animals

Nine-week-old C57BL/6 J male (*n* = 13) and female (*n* = 12) mice housed in rooms with controlled temperature (21–23 °C) and humidity (60–70%), on a 12 h/12 h light/dark cycle, with *ad libitum* access to food and tap water, were used for this study. All procedures regarding handling and care of the animals were in accordance with the established guidelines and approved by the Committee for experimental Animals Wellbeing of the University of Medicine and Pharmacy of Craiova (protocol code 2.13/29.10.2020). The number of animals used and the suffering caused by the study methodology were minimized as much as possible.

### 2.2. Non-Alcoholic Fatty Liver Disease/Non-Alcoholic Steatohepatitis Induction

After relocation to the experimental rooms, mice were given 3 days to acclimatize to the new laboratory conditions. Following the initial 3 days, normal food was replaced with pelleted lacking methionine/choline chloride food (MCD) (MP Biomedicals, Germany) Non-alcoholic fatty liver disease/non-alcoholic steatohepatitis induction [34,35,36], which we used as a non-alcoholic, non-viral hepatitis model (exact food composition Appendix A). After two weeks of MCD food intake, mice (*n* = 20) were randomly divided into two groups (*n* = 10) in each group, with an equal number of males and females per group). All MCD animals received intraperitoneal injections, starting with week 2 (W2), daily as follows: the control group (MCG) received treatment with saline and the treatment group (MCG+Cy) was given 10 mg/kg Cerebrolysin (Ever Pharma, Oberburgau, Austria). A number of five animals were kept as sham witch were fed normal food and received intraperitoneal injections of saline starting with W2 of the experiment. All the mice were fed MCD *ad libitum* for 4 weeks, until they were 14 weeks old. During the experimental period a number of 3 animals (Sham = 1 and MCG = 2 died during the anesthesia/ultrasonography procedure).

### 2.3. Clinical Evaluation and Behavior Testing

Mice were inspected daily for distress and their body mass was measured weekly, for the entire duration of the experiment. Food intake was measured daily also and all unconsumed food from the precedent day was removed and replaced. All mice were tested before the animals were fed methionine/choline chloride diet (week 0–W0), right before the initiation of the treatment (W2), and before tissue harvesting (week 4–W4). Before each test, all surfaces were wiped with 75% ethanol to remove odors.

Open field (OF) test: Mice were put in the open field maze, which measured 50 cm (length) × 33 cm (width) × 15 cm (height). They were allowed to move and explore freely for 10 min. In each test, a single mouse was put in the center of the arena. All trials were recorded (USB webcam) and analyzed (EthoVision XT 14, Noldus Technology). The total exploration time (time in the center area (s)), total distance (cm) and average speed (cm/s) were determined.

Novel object recognition (NOR) test: At the beginning of the test, the mice could freely explore two identical objects placed at a distance of 15 cm from the side walls in two opposite corners of the apparatus for 6 min. After 60 min, one of the objects was replaced by a novel one, and the tested mouse was allowed to explore again. Preference index was defined as the percentage of the time exploring one identical object within the total time exploring both objects. Recognition index was defined as the percentage of the time exploring the novel object among the total time of exploring both objects.

### 2.4. Abdominal Ultrasonography

Initially, the mice underwent several liver parameters and vessels measurements using the ultrasonography technique at the beginning of the study, in the 2nd and 4th week. A L40-8/12 plane probe, for soft tissue, from Ultrasonix Sonix Touch Ultrasound was used. After light intraperitoneal anesthesia using 30 µL of a ketamine (120 mg/kg) and xylazine (10 mg/kg) solution, the animals were placed on their backs and their fur was removed from the upper area of the abdomen. Right and left portion of the median liver lobes, diameters of the aorta, inferior vena cava and portal vein were manually measured by para medio sagittal and transverse sections (4 sections/animal) using the inbuilt ruler of the machine for each section best visualizing the investigated structure. The same parameters were used in the ultrasound settings for the measurements: frequency 2.0 MHz, depth 1.3 cm, gain 55%. Severity score was marked following parenchymal echotexture, nodule presence and surface of margin of the liver (see Appendix A for details)

### 2.5. Histopathology and Immunohistochemistry

Brain and liver tissue were harvested from anesthetized mice that were intracardially perfused with saline 5 mL saline, followed by 5 mL 4% paraformaldehyde and overnight fixation as this method has been proven to have the least impact on microglial activation [37]. After fixation, some of the tissue was included in paraffin while other was frozen. The liver samples were deparaffined, rehydrated and further stained with Hematoxylin and Eosin, Periodic-Acid-Schiff and Masson Trichrome staining. The hepatic lesion was quantified according to a previous histological scoring for nonalcoholic fatty liver model (see Appendix A for details) [38]. For chemical immunohistochemistry, sections were further processed for antigen retrieval in citrate buffer (0.1 M, pH 6) by microwaving them for 20 min, at 650 W. After cooling to room temperature, endogenous peroxidase was inhibited with a 1% solution of water peroxide for 30 min, then the unspecific binding sites were blocked in 3% skimmed milk (Biorad, Hercules, CA, USA) for another 30 min. For enzymatic detection, the slides were incubated (18 h) at 4 °C, with the primary antibodies (monoclonal Anti-NeuN mouse at a 1:1.000, MilliporeSigma, St. Louis, MO, USA). Next day, the signal was amplified with a species-specific peroxidase labelled polymer (Nichirei Biosciences, Tokyo, Japan) for 1 h, and visualized with 3,3′-diaminobenzidine (Nichirei Biosciences, Tokyo, Japan). After hematoxylin counterstaining, slides were covered with xylene based mounting medium (Sigma-Aldrich, St. Louis, MO, USA). Images were collected randomly from the cortex regions of interest (primary somatosensory cortex), capturing at least three microscopic fields (671.38 × 562.95 µm) for each region.

Fluorescence immunohistochemistry. Following freezing procedures, 25 μm thick sections were cut and prepared for immunohistochemistry following standard protocols, as previously described [39]. Antibodies against Iba1 (1:3000, Wako Chemicals USA Inc., Richmond, VA, USA) were used to identify macrophages and microglia. Negative controls that omit primary antibodies and positive controls were applied for each case. The positive cells were counted using a 40× magnification in the same cortical region as in the case of chemical IHC. For 3 consecutive days, after the start of the treatment (W2) all animals received two intraperitoneal injections of 5-bromo-20-deoxyuridine (BrdU, 50 mg/kg; Sigma-Aldrich, St. Louis, MO, USA) (8:00 and 20:00). The acquired images were furthered quantified: the number of NeuN^+^ cells were manually counted, while the area of BrdU and Iba1 signal was quantified using Fiji [40].

### 2.6. Statistical Analysis

Statistical analysis was performed using GraphPad 9.2 and Microsoft Excel 2016. Immunohistochemistry results were evaluated multiple comparison two-way ANOVA (Tukey’s multiple comparisons test) after the data set passed normality testing. Unless noted otherwise, all figures show mean value and standard deviation (SD) and the statistical significance is displayed as follows: * *p* < 0.05, ** *p* < 0.01, *** *p* < 0.001 and **** *p* < 0.0001.

## 3. Results

### 3.1. Cerebrolysyne Does Not Worsened Hepatic Damage Induced by MCD Food

The ultrasonography performed on the animals was able to discriminate between the Sham group and MCG group. With the animals in the Sham having no hepatic ultrasound changes during the experiment, compared to W0 (Figure 1a,a’), the MCD groups started to exhibit hepatic ultrasonography changes, starting with week 2 (Figure 1b,b’) that were gradually getting worse until W4. At this point micro and macro nodules were seen in most of MCD animals (arrows in Figure 1c,c’). This change were quantified the severity score (Figure 1d). Although, some individual changes were dramatic, overall, MCG animals exhibited no changes in the diameter of the right hepatic lobe compared to Sham (Figure 1e). However, at W2 the left hepatic lobe (Figure 1f) was smaller (4.63 ± 1.49 mm compared to 8.79 ± 1.29 mm; *p* < 0.001). The measurements of the large abdominal vessels revealed that the MCG animals suffered no change in the diameter of the aorta (Figure 1g), but starting with W2 the diameter of the inferior vena cava was smaller (1.44 ± 0.22 mm) compared to Sham (1.9 ± 0.37) (Figure 1h) (*p* = 0.006 at W2 and 0.004 at W4). No changes were observed in the diameter of the portal vain (Figure 1i). The administration of Cerebrolysin had no effect on the ultrasonography parameters, with all changes observed in the MCG also seen in the MCG+Cy animals. Animals in both groups lost a comparable body weight (Appendix A), with all animals fed MCD consuming less food starting with W2 compared to Sham. The Cerebrolysin treatment generated a higher food intake (3.73 ± 0.42 g/day) compared to MCG (3.09 ± 0.57) at W4 (Appendix A) (*p* = 0.011).

The liver histological severity scoring (see Appendix A for details), showed no change in the liver architecture of Sham animals. However, in the MCG steatosis was frequently seen (Figure 2a, stars) with intra-lobular discreet to moderate diffuse chronic inflammation (Figure 2b, arrows). The MCG presented numerous microglanulomas (Figure 2c) in which macrophages, lymphocytes and rare neutrophils could be seen. Periportal chronic inflammatory infiltrate could also be noted, spilling on occasion through the marginal hepatocyte cords (Figure 2d, arrows). The Cerebolysin treatment had no influence over the presence of steatosis (Figure 2e), microgranulomas (Figure 2g) or periportal inflammation (Figure 2h), however, intra-lobular inflammation was more frequently seen in MCG compared to MCG+Cy (Figure 2f). Other histological inflammatory signs such as lipogranulomas and mild periportal fibrosis could rarely be seen in both MCG and MCG+Cy.

### 3.2. Cerebrolysine Does Not Improve Bevaivior in Animals Feed MCD Food

Behavior testing performed before and after the initiation of the treatment showed that, after two weeks of MCD food, Sham (Figure 3a) and MCG (Figure 3b) had no difference in the velocity of their movement within the OF arena (Figure 3d) and spent similar times exploring it (Figure 3f). However, the total distance that the MCG animals was lower compared to Sham (2987.1 ± 432.07 compared to 4154.47 ± 855.87 cm, *p* = 0.002) (Figure 3e). After four weeks of MCD food, the MCG group showed lower OF performance compared to Sham, with a decreased velocity (Figure 3d), total distance (Figure 3e) and timed spent exploring the center of the arena (Figure 3f). The two-week treatment daily intraperitoneal Cerebrolysin treatment had no effect on the velocity and total distance, but MCG+Cy animals were less inclined in exploring the center of the arena compared to MCG (66.97 ± 34.97 s compared to 123 ± 70.5 s, *p* = 0.039)

Testing short-term memory after the animals were fed MCD food for 4 weeks showed a highly altered behavior for the sham group, with almost all animals exploring exclusively one object, regardless of its status (Figure 3g). The calculated D2 index showed that the Sham recognized the new object and spent more time exploring it (D2 = 0.59 ± 0.14) while at W2, the MCG had a D2 of 0.25 ± 0.38 and the MCG+Cy had 0.01 ± 1 (*p* > 0.05).

### 3.3. Cerebrolysin Improves Neuronal Survivability but Has No Impact on Neuroinflammation

In order to analyze the cellular response to 4 weeks of MCD food intake, and how the Cerebrolysin could influence it, we decided to investigate both neuronal and inflammatory response. We observed that after the 4 weeks, in the MCG group, the number of NeuN^+^ cells dropped to 173,092 ± 10,057/mm^3^ from a mean of 196,632 ± 10,378/mm^3^ in Sham (Figure 4a)(*p* = 0.006). No difference between Sham and MCG was observed when analyzing the BrdU (Figure 4b) or Iba1 signal (Figure 4c). In the treated group we could observe that the number of NeuN^+^ cells did not decrease, as in the case of MCG (Figure 4a). The treated animals had more BrdU signal (3872 ± 1856 µm^2^) compared to both MCG (2143 ± 856 µm^2^) and Sham (154.1 ± 42.24 µm^2^) (Figure 4b). We did not observe and increase in the area occupied by Iba1^+^ cells (Figure 4c) in the investigated region of interest.

## 4. Discussion

Although mainly regarded as an organ specific disease, studies have shown that patients suffering from hepatitis can also develop dysfunctions of other systems [41]. Within the so-called secondary organ damage caused by hepatitis, brain alteration are predictors of a negative outcome. For example, patients that develop Hepatic Encephalopathy (HE) are more likely to die compared with patients without it [4]. Although it is not part of the model for end-stage liver disease score, HE is often used in deciding priority of liver grafts [42], and while LT removes the underlying chronic liver dysfunction, patients that undergo grafting frequently present a certain cognitive impairment even after the procedure [9,10,11,12].

We are not the only group that observed the lack of understanding the pathological mechanisms underlying cognitive impairment seen in patients after LT and that there is no clear consensus of the nomenclature [43].

Depending on the initial cause of hepatitis, patients can have different risks to develop neurological symptoms. While up to 50% of patients with hepatitis C virus can often report peripheral neuropathy (mainly sensory), cognitive impairment, cerebrovascular accidents, encephalitis, myelitis, encephalomyelitis or Guillain-Barré syndrome [8,9], patients with hepatitis B rarely develop neurological disorders [44].

Considering the high frequency and health implication of viral hepatitis (neurological ones included), it is understandable why the vast majority of clinical studies have focused on this type of liver inflammation. In contrast, non-viral hepatitis does not receive the same attention. This lack of fundamental and clinical data regarding non-viral hepatitis is curious as its complications such as liver failure (seen in advanced stages of cirrhosis), increased gut permeability (that generates an over production of multiple inflammatory cytokines) can contribute to blood-brain barrier permeability alteration leading to HE [45]. Because patients suffering from HE can present different degrease of cognitive dysfunction from mild impairment, to coma and even death [46], sometimes the correct identification of this pathology is delayed [47], contributing to the high morality within this group [4,48].

The understanding of the physio-pathological processes can lead to new treatment strategies. Because of the promising results of some neuroprotective treatments in the experimental setup [24,26] and clinical practice [25,49], we investigated if this kind of treatment could also help protect patients suffering from liver damage. Although the treatment did not yield a huge behavior change, with both treated and untreated groups being less mobile (Figure 3d,e) compared to sham and no apparent short-term memory benefit being observed (Figure 3g), we did see some cellular benefits to the treatment. With the average number of NeuN^+^ cells in the treated animals being comparable to sham and surprisingly in increase in the area of BrdU^+^ cells, the treatment seems to have a positive impact on the cells of the cortex. No difference was observed when analyzing the area occupied by Iba1^+^ cells, showing that the effect of the treatment does not affect the inflammatory state of the brain associated with liver damage [42,43]. This was surprising as previous studies done on rats, show a decrease of neuroinflammation after Cerebrolysin treatment [50]. This difference could be attributed to the fact that inflammation was evaluated on cell cultures stimulated with lipopolysaccharide, while here the BBB wasn’t as damaged as to generate a huge immune response of microglia/macrophages. This might be a consequence of the relatively acute nature of the animal model, since the animals developed steatosis or steato-hepatitis, as also described by other studies [34,51] but they were not subjected to a longer period of time to live with this condition. Although performed over a relatively short period of time, impaired liver function was achieved in our experiment because a well-established experimental model was used [52,53]. The presence of microgranulomas (Figure 2c) and lipogranulomas (Figure 2d) are additional indicators of liver damage [54,55]. The ultrasound examination confirmed the architectural changes in the liver starting with W2, which intensified until W4 (Figure 1a–c). In the clinic setting, a patient suspected of a secondary brain lesion due to liver failure will be subjected to a plethora of clinical tests starting with mental status evaluation or cognitive functions assessment all the way to a complete neurological examination (including deep-tendon reflexes and cerebellar signs). From here, such a patient could also undergo biochemical diagnostic or neurophysiological assessment. All those tests should generate a complete picture of the patient brain function. Under our experimental condition, such a detailed examination will be impractical. This can explain why our behavior testing was not able to identify differences between the MCG and MCG+Cy groups (Figure 3). Never the less, we were able to show that, at least at a cellular level, a neuroprotective strategy does have the potential to improve cortical outcome in patients with liver disease.

## 5. Conclusions

With deaths from cirrhosis and hepatocellular carcinoma on the rise [56], more and more patients will be needing LT. With neurological complications of end-stage liver disease affecting the rehabilitation of patients receiving liver grafts, it makes sense that, protecting the brain for additional damage while waiting for the procedure, can have a beneficial effect. With some neuroprotective strategies already showing promising results in experimental end-clinical studies, we showed that the same strategies could protect cerebral cells from diffuse damage.

## Figures and Tables

**Figure 1 brainsci-11-01622-f001:**
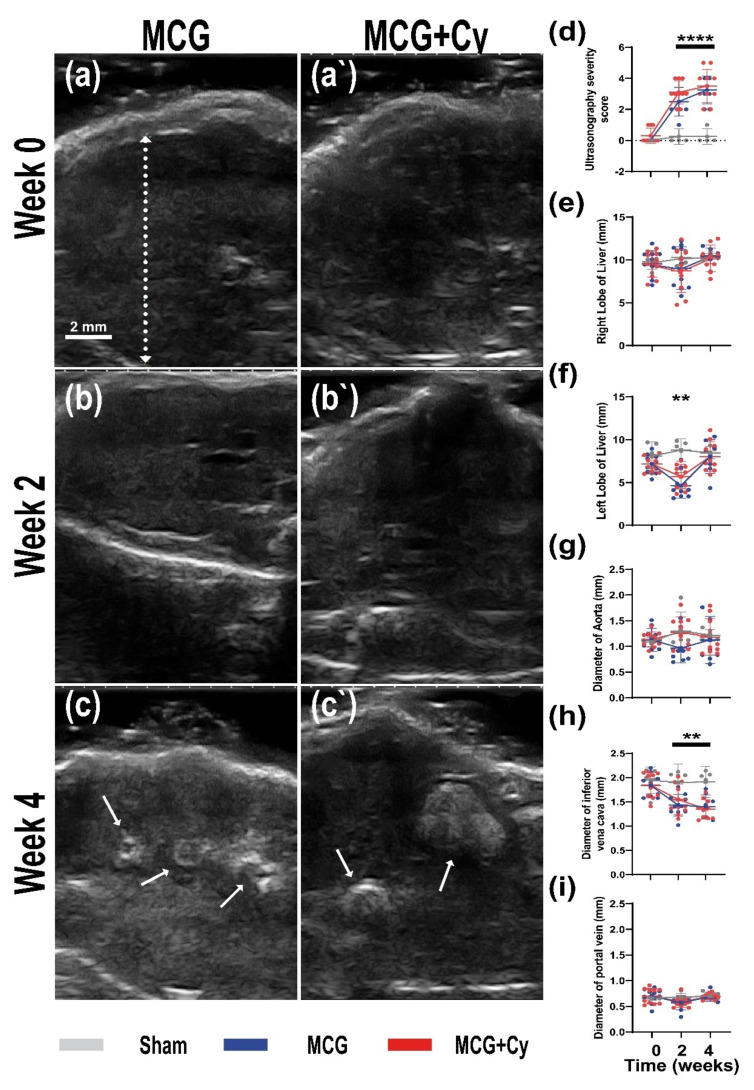
Repetitive hepatic ultrasonography in mice. (**a**,**a’**) During the course of the experiment, hepatic architecture was evaluated using ultrasonography at the beginning (W0), two weeks (W2) after the start of MCD food (**b**,**b’**) and (**c**,**c’**) 4 weeks after (W4). At W4, in all MCD groups (MCG and MCG+Cy), ultrasonography was able to distinguish micro and macro nodules (arrows). (**d**) Compared to Sham, the ultrasonography severity scoring was able to reveal changes in the MCD, starting with W2 (*p* < 0.0001), with the MCG reaching, at W4, an average of 3.25 ± 0.88 in MCG and a 3.5 ± 1.08 in the MCG+Cy, compared to 0.25 ± 0.5 in Sham. Measuring the diameter of the (**e**) right (dotted line in panel (**a**,**f**)) left liver lob revealed that both MCD groups groups had a smaller left liver lob (MCG 4.63 ± 1.49 mm; MCG+Cy 5.69 ± 1.42 mm) compared to 8.79 ± 1.29 mm in Sham (*p* < 0.001). No changes in the diameter of (**g**) the aorta or (**i**) portal vine were observed. (**h**) However, starting with W2, the MCD groups had smaller diameter of inferior vena cava (MCG 1.44 ± 0.22 mm and MCG+Cy 1.54 ± 0.32 mm) compared to Sham (1.9 ± 0.37 mm). Scale bar 2 mm. The graphs show mean values ± SD, ** *p* < 0.01, **** *p* < 0.0001.

**Figure 2 brainsci-11-01622-f002:**
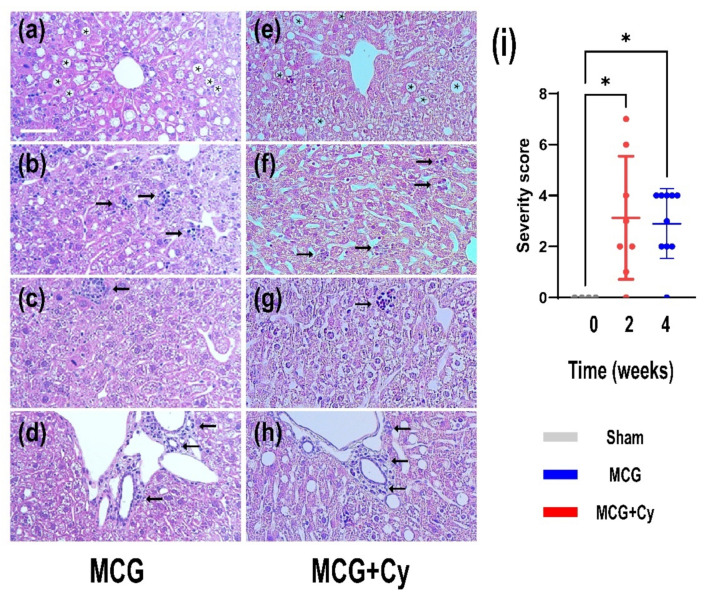
Histological effects of MCD food and Cerebrolysin treatment on the liver of mice. (**a**) steatosis, (**b**) intra-lobular discreet to moderate diffuse chronic inflammation, (**c**) microglanulomas and (**d**) periportal inflammation are present in MCG and (**e**–**h**) MCG+Cy groups. (**i**) This changes have generated a histological severity score of 3.125 ± 2.416 for the MCG group and of 2.9 ± 1.370 of the MCG+Cy group (*p* > 0.05). Scale bar 100 µm. The graphs show mean values ± SD, * *p* < 0.05.

**Figure 3 brainsci-11-01622-f003:**
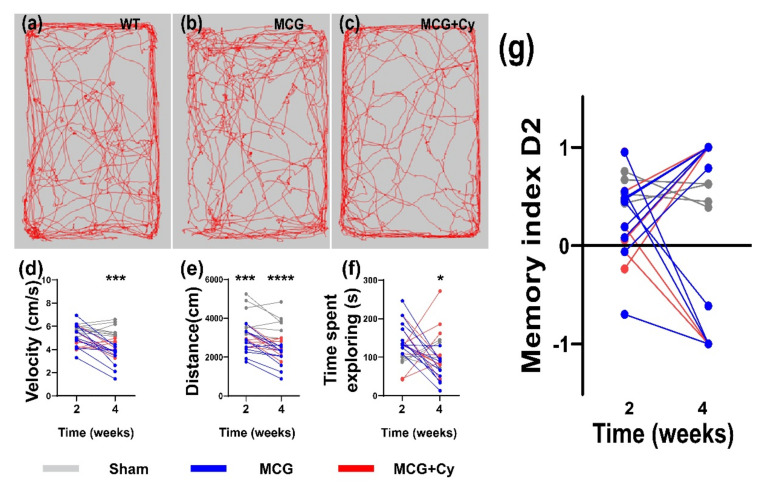
Behavior testing in MCD mice. Open field of (**a**) Sham,(**b**) MCG and (**c**) MCG+Cy mice reveals that at W4 compared to (**e**) Sham (5.71 ± 0.57 cm/s), MCG mice move slower (4.11 ± 0.60 cm/s), *p* = 0.0007, with the MCG+Cy groups moving at an average speed of 3.36 ± 0.97 (*p* < 0.0001) (**e**) and covered less distance (MCG, *p* = 0.0005, and MCG+Cy, *p* < 0.0001). (**f**) When looking at time spent exploring the center of the arena, we found a small difference between the MCG and MCG+Cy groups (*p* = 0.0396). (**g**) NOR testing revealed no difference two weeks and 4 weeks after MCD food intake compared to Sham. The graphs show mean values ± SD, * *p* < 0.05, *** *p* < 0.001 and **** *p* < 0.0001.

**Figure 4 brainsci-11-01622-f004:**
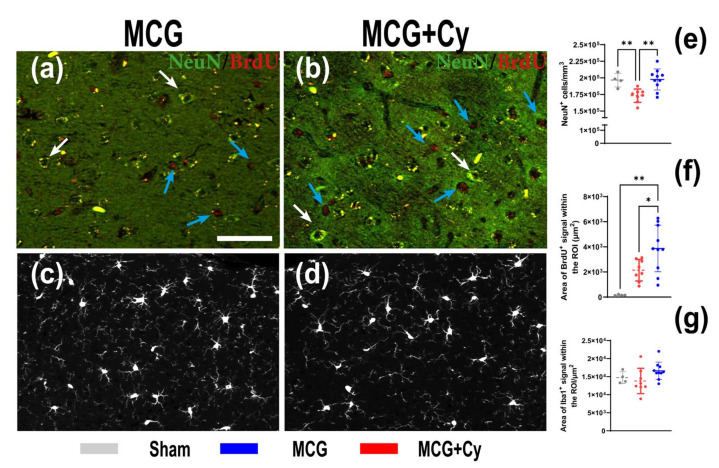
Cortical cellular response to Cerebrolysin in mice feed MCD. (**a**,**e**) MCG mice had less neurons after 4 week of MCD compared to Sham (173,092 ± 10,057/mm^3^ vs. 196,632 ± 10,378/mm^3^
*p* = 0.006) food (white arrows). (**b**,**e**) 2 weeks of Cerebrolysin (10 mg/kg) prevented this drop (197,555 ± 15,876/mm^3^, *p* > 0.05). (**a**,**f**) The BrdU area in MCG although has increased to 2143 ± 856 µm^2^ compared to 154.1 ± 42.24 µm^2^ in Sham it did not reach significance (*p* = 0.07) (red arrows). In the MCG+Cy group, the BrdU area was higher than both Sham (*p* = 0.001) and MCG (*p* = 0.02). Iba1 staining in (**c**) MCG and (**d**) MCG+Cy (**g**) revealed no difference compared to Sham. Scale bar 50 µm. The graphs show mean values ± SD, * *p* < 0.05, ** *p* < 0.01.

## Data Availability

All data is available upon request.

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
