# Peer review of "Cerebrolysin Prevents Brain Injury in a Mouse Model of Liver Damage"

_brainsci, 2021, doi:10.3390/brainsci11121622_

Round 1
Reviewer 1 Report
One of the silent epidemics that run in the past decades is the one of non-alcoholic steatohepatitis. As this condition is silent, it remains severely underdiagnosed globally and special efforts for raising awareness are necessary. This is why the current paper presents experimental work in a very significant area and deserves to be published although after certain revisions. Several recommendations of mine are listed below and in summary I believe that the article needs to be revised in order to include better description of the background of the problem, pathogenesis of hepatic encephalopathy, better description of the experimental design and results.
Introduction
Pathogenesis of hepatic encephalopathy and potential approaches for its alleviation/treatment needs to be included in order to rationalize the experimental treatment Cerebrolysine.
Materials and methods
- How was the saline administered in the control group? Was the control group subjected to intraperitoneal invasion as well?
- What was the sham group subjected to? Fed normal food? Got saline i.p. injection?
Better description of the groups and the procedures they have been subjected to and the experimental design in general is necessary.
Reference to the animal model used is missing.
Methionine/choline-DEFICIENT diet is a classical dietary model for studying Non-alcoholic steatohepatitis. From what we read in the paper it looks like the animals were fed with methionine/choline chloride supplemented food: “pelleted methionine/choline chloride food (MP Biomedicals, Germany)” sounds like supplemented food. This needs to be clarified.
The expected level of protection by Cytolysine and the expected differences in the MCG+CY and MCG control group could have been better captured with more comprehensive neuropsychological testing including tests not only for short term memory but also spatial memory (variety of maze tests), motor coordination (Rotarod test), learning and memory (active or passive avoidance). After all the primary endpoint of this experiment is to show changes in the neuropsychological status.
Further, the model itself provides a relatively mild chronic liver condition (NASH). With the condition being chronic, authors could have assessed several timepoints and not just 1.
Author Response
One of the silent epidemics that run in the past decades is the one of non-alcoholic steatohepatitis. As this condition is silent, it remains severely underdiagnosed globally and special efforts for raising awareness are necessary. This is why the current paper presents experimental work in a very significant area and deserves to be published although after certain revisions. Several recommendations of mine are listed below and in summary I believe that the article needs to be revised in order to include better description of the background of the problem, pathogenesis of hepatic encephalopathy, better description of the experimental design and results.
We thank the reviewer for his appreciations and insights.
Introduction
Pathogenesis of hepatic encephalopathy and potential approaches for its alleviation/treatment needs to be included in order to rationalize the experimental treatment Cerebrolysine.
We will like to thank the reviewer for suggesting these changes. We think it that the rewritten introduction will better fit the purpose of this article. As such, it now reads as:
“Liver damage is an important cause of mortality and morbidity worldwide. In 2016, cirrhosis was reported to be the 11th leading cause of death and 15th cause of morbidity [1]. Since the liver plays a key role in detoxification, liver dysfunction, regardless of the cause, leads to an accumulation of certain substances and catabolic products [2]. With this increase, a wide range of neurological and neuropsychiatric manifestations can occur [3,4]. Clinical and experimental data have shown that cortical changes are observed in all forms of liver damage, but it is still unclear which cellular and molecular signaling pathways are affected and at what extend. Since around approximately 30% of normo-ponderal and almost 80% of all overweight people are affected by non-alcoholic fatty liver disease/non-alcoholic steatohepatitis (NAFLD/NASH), this has developed into a silent epidemics in the last few decades [5-7]. With neurological and neuropsychiatric manifestations that vary from acute manifestations (encephalitis, myelitis, encephalomyelitis or Guillain-Barré syndrome [8,9],) to chronic manifestations such as cognitive impairment and dementia [10,11], the exact molecular pathways of this alterations are not completely understood. Theories explaining cortical changes are incriminating the altered permeability of blood-brain barrier (BBB) which allow a large number of small polar molecules to enter the brain, ultimately leading to edema [12-15]. However, insulin resistance, inflammation, hormonal alterations and change in levels of secreted hepatokines [16] have also been reported in this patients.
Some of them will ultimately need a liver transplantation (LT), since grafting is the only definitive treatment for end-stage liver disease, independent of the ethology [17,18]. After transplantation, the BBB permeability impairment can normalize. Regardless, the already acquired brain damage can be permanent. Since 10 to 85% of LT recipients experience central nervous system (CNS) complications (ranging from focal to diffuse brain injury), the main issue with this types of patients is that CNS damage can impair both the short and long-term outcome [19]. In addition, pretreatment with calcineurin inhibitors drugs may exacerbate dysarthria, akinetic mutism, confusion and seizures [20-22].
Given all evidence of cortical changes in patients with liver damage [5,23] and the long-term implication of such changes, a neuroprotective treatment may be considered. Clinical and experimental studies have shown that in other neurological condition like stroke [24], some encephalopathies [25], especially experimental autoimmune encephalomyelitis [26], certain neuroprotective strategies, such as using a Cerebrolysin, a cocktail of low molecular weight peptides and amino acids [27,28]. As the exact mechanisms of its neuroprotective and neurotrophic effects are heterogenous. Reports have attributed its neurotrophic effect to imitations, seemingly through activating the e PI3K/Akt pathway, in a similar manner to o brain derived neurotrophic factor (BDNF) [29-31]. Cerebrolysin's neuroprotective effects have been largely attributed to its role in activating the Sonic Hedgehog signaling pathway [32], but reports have also shown that it is able to lower free radical levels in the cortex and reduce the amount of proapoptotic enzymes [33]. With multiple mechanism involved in the cerebral alterations of liver damage, a single molecular pathway approach for neuroprotection seems not appear to be feasible, but with multiple ways in which Cerebrolysin increases neuroprotection, it seems like a perfect candidate for the task be.”
Materials and methods
- How was the saline administered in the control group? Was the control group subjected to intraperitoneal invasion as well?
- What was the sham group subjected to? Fed normal food? Got saline i.p. injection?
Better description of the groups and the procedures they have been subjected to and the experimental design in general is necessary.
Reference to the animal model used is missing.
Methionine/choline-DEFICIENT diet is a classical dietary model for studying Non-alcoholic steatohepatitis. From what we read in the paper it looks like the animals were fed with methionine/choline chloride supplemented food: “pelleted methionine/choline chloride food (MP Biomedicals, Germany)” sounds like supplemented food. This needs to be clarified.
We thank the reviewer for his comments. We agree that the description of the experimental was a bit ambiguous. We rewritten it and it now reads as:
”After relocation to the experimental rooms, mice were given 3 days to acclimatize to the new laboratory conditions. Following the initial 3 days, normal food was replaced with pelleted lacking methionine/choline chloride food (MCD) (MP Biomedicals, Germany) to induce a non-alcoholic steatohepatitis (NASH) [20-22] which we used as a non-alcoholic, non-viral hepatitis model. After two weeks of MCD food intake, mice (n=20) were randomly divided into two groups (n = 10 in each group, with an equal number of males and females per group). All MCD animals received intraperitoneal injections, starting with W2, daily as follows: the control group (MCG) received treatment with serum saline and the treatment group (MCG+Cy) that was given daily intraperitoneal injections with10 mg/kg Cerebrolysin (10mg/kg) (Ever Pharma, Austria). A number of five animals were kept as sham witch were feed normal food and received, intraperitoneal injections of saline starting with week 2 of the experiment. All the mice were fed methionine/choline chlorideMCD ad libitum for 4 weeks, until they were 14 weeks old. During the experimental period a number of 3 animals (sham=1 and MCG=2 died during the anesthesia/ultrasonography procedure).”
The expected level of protection by Cytolysine and the expected differences in the MCG+CY and MCG control group could have been better captured with more comprehensive neuropsychological testing including tests not only for short term memory but also spatial memory (variety of maze tests), motor coordination (Rotarod test), learning and memory (active or passive avoidance). After all the primary endpoint of this experiment is to show changes in the neuropsychological status.
Further, the model itself provides a relatively mild chronic liver condition (NASH). With the condition being chronic, authors could have assessed several timepoints and not just 1.
We will like to thank the reviewer for his comments. It will be an excellent point to argue in the UMFCV animal Wellbeing Committee in future experiments. The reviewer is correct in criticizing the small battery of behavior tests. At the start of the experiment W0, the animals were also tested for long term memory (water maze) and Rotarod. However, at W2, the body weight of the animals had decreasing so much we needed to consult with the local veterinary. His suggestion was not to perform these two tests due to concern for the animals. Also at his suggestion, the experiment was stopped at 4 weeks and we were not able to perform measurements at week 6, as was the initial procedure (not approved by the Committee).
Reviewer 2 Report
The manuscript from Morega and colleagues addresses the impact of the neuropeptide drug Cerebrolysin on the pathology associated to a methionine/choline chloride diet on mice. The authors started by inducing the non-viral, non-alcoholic fatty liver experimental model. They report an altered exploratory drive, motor ability, and recognition memory behaviour on mice under the methionine/choline chloride diet compared to sham mice. Importantly, 2 weeks of Cerebrolysin treatment appears to increase neuronal survival of mice fed this diet regiment, with no apparent beneficial impact in mice short-term memory or brain-microglia/macrophages inflammatory status.
The findings are very interesting and experiments are well performed. But the interpretation of the data is sometimes not sufficiently supported by data or not presented appropriately. Although appropriate, the manuscript needs careful editing to highlight the data presented.
Major Points:
- Methods: It is not clear how the authors induced the non-viral, non-alcoholic fatty liver model in the mice, was it via a methionine/choline chloride deficient diet? This is described as a classical and well established dietary model of NASH/NAFLD. Moreover, a methionine/choline chloride diet is considered normal. Please clarify this issue.
- Results:
Supplementary Figure 1: authors should add side by side the plot showing not only mice body-weight, but also mice food intake throughout the experiment. On the caption, authors wrote “The animals feed methionine/choline chloride lost body mass in the first two weeks of the experiment compared to Sham (p<0.01), but, probably due to the I.P. injection stress, sham started to lose weight in the last part of the experiment”. Again, this is issue is clearly understood if the mice were fed a methionine/choline chloride deficient diet.
Figure 1- Could the authors provide clear pictures from the ultrasonography? Which group is represented in Figures (a-c)? Always make comparisons to the corresponding controls (MCG vs MCG+Cy vs sham). It would be helpful if the authors use signs (arrows with different colours, for instance) in the ultrasonography pictures to highlight the measurement related to figures (e), (f), (g), (h) and (i). Authors must detail how the measurements (Figure (d-i) were done in the Material and Methods section. Finally, authors must clearly discriminate the observed liver pathology between MCG vs Sham and, only afterwards, the effect of Cerebrolysin on the MCG (and sham) to avoid confusion.
Figure 2- The authors only provide figures related to MCG. However, the goal of this manuscript is to study the potential neuroprotective role of Cerebrolysin. This picture only depicts the effect of the methionine/choline chloride diet on liver function. Similarly to Figure 1, authors must clearly discriminate the effect of the diet vs the effect of Cerebrolysin on the MCG to avoid ambiguity. ). It would be helpful if the authors use signs in the pictures to highlight the observation in Figure 2 (a-f).
Figure 3- In the same way to Figures 1-2, authors must clearly discriminate the effect of the diet vs the effect of Cerebrolysin on the MCG to avoid ambiguity.
Figure 4- Authors reported a significant decrease on NeuN+ cells on the MCG, 2 weeks after starting the diet. Notably, Cerebrolysin restored NeuN+ cells to numbers similar to sham group. Thus corroborating the neurotrophic and neurorecovery functions of this drug observed in other neurological conditions such stroke or TBI (Zhang Li et al. Stroke. 2013; Jin Yoming et al. Stroke. 2017; Chen Honghui et al. Neurobiology of aging. 2007; Zhang Chunling et al. J Neurosc Research. 2010). It would be of main relevance if the authors could provide a mechanistic pathway(s) underlying the Cerebrolysin way of action. Also, these observations need further discussion.
Although the authors did not observed any effect of Cerebrolysin on IBA1+ cells there are evidences, on the literature, that this drug might induce neuroprotection by reducing microglia activation and neuroinflammation (Alvarez, XA et al. J Neural Transm Suppl. 2000). Could the authors argue and discuss this topic? Note that the most widely used microglial markers (such as IBA1 or CX3CR1) can be detected irrespective of the cell phenotype. Nowadays, the most specific general microglia markers are transmembrane protein 119 (TMEM119) and purinergic receptor P2Y12R. Thus, the authors should specifically use such markers to show the impact of both methionine/choline chloride diet and Cerebrolysin on microglia status. Please, provide representative IH/IF of the pictures from each group related to Figure 4 (a-c). Moreover, it is not understandable how the authors performed the measurement shown on these; please provide additional details on Material and Methods section.
- Captions should not contain explanation of the results since these are already described in the main text. Conversely, they should include a short title followed by all the elements used to distinguish different groups (experimental conditions) of data on the graph (i.e.,time-points, N, stats test and significance, treatments, controls, methodology, magnification, etc.). All captions must be edited (including in the supplementary data).
- Figures: authors must cite in the main text all the figures within a figure panel (a, b, c, d, etc.). Please review and edit them accordingly.
- (Sub)titles: are a mere question without giving the answer, not the actual results. All (sub)titles in the Results section must be edited (including in the supplementary data).
Minor Points:
- Abstract “can be up to 200 words long” (yours has 252) check MDPI guidelines (https://www.mdpi.com/authors/layout#_bookmark2).
- Introduction: The purpose of the work is not properly introduced. For instances, authors only reflect on the health implications of viral vs non-viral hepatitis issues on “Discussion”. This problematic should be properly introduced in the Introduction section. Moreover, the authors write about treatment strategies proven to be neuroprotective in several disease models, however they never refer to the “Cerebrolysin” treatment specifically. However, the works that are cited refer to this drug. So, “Cerebrolysin” should be properly introduced and as well as the reason(s) why the authors chose this treatment. Please, clearly refer to “Cerebrolysin” treatment rather “the treatment” or “neurotrophic treatment” whenever is relevant.
Line 61: in the sentence: so called “prolonged state of neuroinflammation”, do the authors refer to microglia chronic activation that induces/sustains neuroinflammation? Please review the accuracy of this statement.
- Material and Methods:
Line 81: authors should be consistent when writing, for instances, “non-viral, non-alcoholic hepatitis” – this is not how it was introduced or discussed along the manuscript.
The content of the methionine/choline chloride food should be added supplementary material.
It is not clear when and for how long does the treatment endure. Does it start at week 2? Did it last for 2 weeks, daily ip injections? Any reference study to follow the protocol? Clarify what is the sham, it is unclear what “treatment with serum means” (is it saline?).
Line 150: in this study, authors used BrdU to follow cell proliferation not cell differentiation (cell proliferation is the process of increasing the cell number while cell differentiation is the process of forming a variety of cell types that have specific functions). Please rephrase the statement.
- Results:
Line 164: it is missing the number of the week.
Please list the pictures as they are described in the text.
- Please carefully proof-read the manuscript to minimize typos, grammatical errors and coherency of the text:
Line 5: comma after affiliation number
Line 19: end stage vs end-stage
Line 23: Cerebrolisin vs Cerebrolysin
Line 44: his vs these
Line 75: ad libitum vs ad libitum
Line 164: lever vs liver
Line 260: Never the less vs Nevertheless
…
- Main title: I would like to suggest to use “Cerebrolysin” instead of “Neuroprotection”? “Cerebrolysin” in actually the main purpose of this work.
- US or British English are acceptable, however authors must be consistent throughout the paper.
- Make sure that all abbreviations are defined and be consistent throughout the paper.
- Please check the guidelines regarding “Appendixes and Supplementary Information”, authors must be consistent whether they choose to add sections as Appendix (labelled as Figure A1 or Table A1) or as Supplementary files.
- Statistical significance: all data that is ∗p < 0.05 is statistical significant. I don’t personally see the purpose of distinguishing ∗∗∗p < 0.001 and ∗∗∗∗p < 0.0001. It is enough ∗p < 0.05 and ∗∗p < 0.01.
- Funding: any grant should be mentioned in this section.
- Acknowledgments: are a place to recognize any contributions made to the paper that do not meet the criteria for authorship.
Author Response
Reviewers 2
The manuscript from Morega and colleagues addresses the impact of the neuropeptide drug Cerebrolysin on the pathology associated to a methionine/choline chloride diet on mice. The authors started by inducing the non-viral, non-alcoholic fatty liver experimental model. They report an altered exploratory drive, motor ability, and recognition memory behaviour on mice under the methionine/choline chloride diet compared to sham mice. Importantly, 2 weeks of Cerebrolysin treatment appears to increase neuronal survival of mice fed this diet regiment, with no apparent beneficial impact in mice short-term memory or brain-microglia/macrophages inflammatory status.
The findings are very interesting and experiments are well performed. But the interpretation of the data is sometimes not sufficiently supported by data or not presented appropriately. Although appropriate, the manuscript needs careful editing to highlight the data presented.
Major Points:
- Methods: It is not clear how the authors induced the non-viral, non-alcoholic fatty liver model in the mice, was it via a methionine/choline chloride deficient diet? This is described as a classical and well established dietary model of NASH/NAFLD. Moreover, a methionine/choline chloride diet is considered normal. Please clarify this issue.
We thank the author for his careful reading of the manuscript and his helpful suggestions. The model used is indeed NASH/NAFLD and we made sure to highlight that in the Material and Method. The section now reads as:
“After relocation to the experimental rooms, mice were given 3 days to acclimatize to the new laboratory conditions. Following the initial 3 days, normal food was replaced with pelleted lacking methionine/choline chloride food (MCD) (MP Biomedicals, Germany) to induce a non-alcoholic steatohepatitis (NASH) [20-22] which we used as a non-alcoholic, non-viral hepatitis model. After two weeks of MCD food intake, mice (n=20) were randomly divided into two groups (n = 10 in each group, with an equal number of males and females per group). All MCD animals received intraperitoneal injections, starting with W2, daily as follows: the control group (MCG) received treatment with serum saline and the treatment group (MCG+Cy) that was given daily intraperitoneal injections with10 mg/kg Cerebrolysin (10mg/kg) (Ever Pharma, Austria). A number of five animals were kept as sham witch were feed normal food and received, intraperitoneal injections of saline starting with week 2 of the experiment. All the mice were fed methionine/choline chloride MCD ad libitum for 4 weeks, until they were 14 weeks old. During the experimental period a number of 3 animals (sham=1 and MCG=2 died during the anesthesia/ultrasonography procedure).”
- Results:
Supplementary Figure 1: authors should add side by side the plot showing not only mice body-weight, but also mice food intake throughout the experiment. On the caption, authors wrote “The animals feed methionine/choline chloride lost body mass in the first two weeks of the experiment compared to Sham (p<0.01), but, probably due to the I.P. injection stress, sham started to lose weight in the last part of the experiment”. Again, this is issue is clearly understood if the mice were fed a methionine/choline chloride deficient diet.
We appreciate the reviewer’s comments and adjusted accordantly.
Figure 1- Could the authors provide clear pictures from the ultrasonography? Which group is represented in Figures (a-c)? Always make comparisons to the corresponding controls (MCG vs MCG+Cy vs sham). It would be helpful if the authors use signs (arrows with different colours, for instance) in the ultrasonography pictures to highlight the measurement related to figures (e), (f), (g), (h) and (i). Authors must detail how the measurements (Figure (d-i) were done in the Material and Methods section.Finally, authors must clearly discriminate the observed liver pathology between MCG vs Sham and, only afterwards, the effect of Cerebrolysin on the MCG (and sham) to avoid confusion.
We added markers to highlight some of the parameters used to make all the measurements. The picture quality is the best that the machine can output, for the dimensions measured. It might be that the upload process also decreased the quality of the pictures, as in the downloaded pdf the pictures are of lower quality that the uploaded ones. We also added the sentence “Right and left portion of the median liver lobes, diameters of the aorta, inferior vena cava and portal vein were manually measured by para medio sagittal and transverse sections (4 sections/animal) using the inbuilt ruler of the machine for each section best visualizing the investigated structure”. In the material and Method section to clear the way the measurements were made.
Unfortunately, we used 4 sections/animal to do all the measurements and putting all of them in a figure will take to mush space and, we feel it does not add to the value of the results. As the reviewer suggested we first commented on the changes between Sham and MCG and after on the effect that Cerebrolysin had.
Figure 2- The authors only provide figures related to MCG. However, the goal of this manuscript is to study the potential neuroprotective role of Cerebrolysin. This picture only depicts the effect of the methionine/choline chloride diet on liver function. Similarly to Figure 1, authors must clearly discriminate the effect of the diet vs the effect of Cerebrolysin on the MCG to avoid ambiguity. ). It would be helpful if the authors use signs in the pictures to highlight the observation in Figure 2 (a-f).
We rewritten the result section regarding Figure 2. We now discussed the alteration of the MCG and Sham first and then the alterations Cerebrolysin had on the liver histology. The section now reads:
” The liver histological severity scoring (see supplementary Table 2 for details), showed no change in the liver architecture of Sham animals. However, in the MCG steatosis was frequently seen (Figure 2a, stars) with intra-lobular discreet to moderate diffuse chronic inflammation (Fig 2b, arrows). The MCG presented numerous microglanulomas (Figure 2c) in which macrophages, lymphocytes and rare neutrophils could be seen. Periportal chronic inflammatory infiltrate could also be noted, spilling on occasion through the marginal hepatocyte cords (Figure 2d, arrows). The Cerebolysin treatment had no influ-ence over the presence of steatosis (Figure 2e), microgramulomas (Figure 2g) or periportal inflammation (Figure 2h), however, intra-lobular inflammation was more frequently seen in MCG compared to MCG+Cy (Figure 2g). Other histological inflammatory signs such as lipogranulomas and mild periportal fibrosis could rarely be seen in both MCG and MCG+Cy.”
Figure 3- In the same way to Figures 1-2, authors must clearly discriminate the effect of the diet vs the effect of Cerebrolysin on the MCG to avoid ambiguity.
The same as in the case of Figure 2, we have rewritten the captions and the text for Figure 3. It now reads as:
“Behavior testing performed before and after the initiation of the treatment showed that, after two weeks of MCD food, Sham (Figure3a) and MCG (Figure 3b) had no differ-ence in the velocity of their movement within the OF arena (Figure 3d) and spent similar times exploring it (Figure 3f). However, the total distance that the MCG animals was lower compared to Sham (2987.1±432.07 compared to 4154.47±855.87 cm, p=0.002) (Figure 3e). After four weeks of MCD food, the MCG group showed lower OF performance compared to Sham, with a decreased velocity (Figure 3d), total distance (Figure 3e) and timed spent exploring the center of the arena (Figure 3f). The two-week treatment daily intraperoneal Cerebrolysin treatment had no effect on the velocity and total distance, but MCG+Cy ani-mals were less inclined in exploring the center of the arena compared to MCG (66.97±34.97s compared to 123±70.5s, p=0.039)
Testing short-term memory after the animals were fed MCD food for 4 weeks showed a highly altered behavior for the sham group, with almost all animals exploring exclu-sively one object, regardless of its status (Figure 3g). The calculated D2 index showed that the Sham recognize the new object and spent more time exploring it (D2=0,59±0.14) while at W2, the MCG had a D2 of 0.25±0.38 and the MCG+Cy had 0.01±1 (p>0.05).”
Figure 4- Authors reported a significant decrease on NeuN+ cells on the MCG, 2 weeks after starting the diet. Notably, Cerebrolysin restored NeuN+ cells to numbers similar to sham group. Thus corroborating the neurotrophic and neurorecovery functions of this drug observed in other neurological conditions such stroke or TBI (Zhang Li et al. Stroke. 2013; Jin Yoming et al. Stroke. 2017; Chen Honghui et al. Neurobiology of aging. 2007; Zhang Chunling et al. J Neurosc Research. 2010). It would be of main relevance if the authors could provide a mechanistic pathway(s) underlying the Cerebrolysin way of action. Also, these observations need further discussion.
We added, in the rewritten introduction, the knowned molecular pathways in which Cerebrolysin is thought to perform his neuroprotective roles and argued why it is a good candidate for neuroprotection in patients with liver damage.
Although the authors did not observed any effect of Cerebrolysin on IBA1+ cells there are evidences, on the literature, that this drug might induce neuroprotection by reducing microglia activation and neuroinflammation (Alvarez, XA et al. J Neural Transm Suppl. 2000). Could the authors argue and discuss this topic? Note that the most widely used microglial markers (such as IBA1 or CX3CR1) can be detected irrespective of the cell phenotype. Nowadays, the most specific general microglia markers are transmembrane protein 119 (TMEM119) and purinergic receptor P2Y12R. Thus, the authors should specifically use such markers to show the impact of both methionine/choline chloride diet and Cerebrolysin on microglia status. Please, provide representative IH/IF of the pictures from each group related to Figure 4 (a-c).
We agree with the reviewer and included the phrase “This was surprising as previous studies done on rats, show a decrease of neuroinflammation after Cerebrolysin treatment [50]. This difference could be attributed to the fact that inflammation was evaluated on cell cultures stimulated with lipopolysaccharide, while here the BBB wasn’t as damaged as to generate a huge immune response of microglia/macrophages.” To discuss the differences from Alvarez et al., 2000.
Regarding the antibodies,, again, the reviewer is correct of pointing the issues with IBA1 staining, however, if neuroinflamation would have accrued, in the context of an altered BBB, the microglia population by its self will not show the additional macrophages that entered the brain.
Plus, in certain pathologies some studies suggest that TMEM119 “is not a stable microglia marker” (https://jneuroinflammation.biomedcentral.com/articles/10.1186/s12974-021-02105-2). As no proper investigation was done on the markers suggested by the reviewer in the context discussed in the manuscript, we opted for a “classical” microglia/macrophage marker.
Moreover, it is not understandable how the authors performed the measurement shown on these; please provide additional details on Material and Methods section.
To clarify we added the sentence:” The acquired images were furthered quantified: the number of NeuN+ cells were manually counted, while the area of BrdU and Iba1 signal was quantified using Fiji [40]” in the Material and Method section.
- Captions should not contain explanation of the results since these are already described in the main text. Conversely, they should include a short title followed by all the elements used to distinguish different groups (experimental conditions) of data on the graph (i.e.,time-points, N, stats test and significance, treatments, controls, methodology, magnification, etc.). All captions must be edited (including in the supplementary data).
We thank the reviewer for his suggestions. We have changed the captions according to his comments.
- Figures: authors must cite in the main text all the figures within a figure panel (a, b, c, d, etc.). Please review and edit them accordingly.
We apopogise for this oversite. We have cited all the figures
- (Sub)titles: are a mere question without giving the answer, not the actual results. All (sub)titles in the Results section must be edited (including in the supplementary data).
We thank the reviewer for his suggestions. We have changed the captions according to his comments.
Minor Points:
- Abstract “can be up to 200 words long” (yours has 252) check MDPI guidelines (https://www.mdpi.com/authors/layout#_bookmark2).
The reviewer is right to point out the abstract requirements. We have rewritten the abstract in order to comply with MDPI guidelines.
- Introduction: The purpose of the work is not properly introduced. For instances, authors only reflect on the health implications of viral vs non-viral hepatitis issues on “Discussion”. This problematic should be properly introduced in the Introduction section. Moreover, the authors write about treatment strategies proven to be neuroprotective in several disease models, however they never refer to the “Cerebrolysin” treatment specifically. However, the works that are cited refer to this drug. So, “Cerebrolysin” should be properly introduced and as well as the reason(s) why the authors chose this treatment. Please, clearly refer to “Cerebrolysin” treatment rather “the treatment” or “neurotrophic treatment” whenever is relevant.
We will like to thank the reviewer for suggesting these changes. We think it that the rewritten introduction will better fit the purpose of this article. As such, it now reads as:
“Liver damage is an important cause of mortality and morbidity worldwide. In 2016, cirrhosis was reported to be the 11th leading cause of death and 15th cause of morbidity [1]. Since the liver plays a key role in detoxification, liver dysfunction, regardless of the cause, leads to an accumulation of certain substances and catabolic products [2]. With this increase, a wide range of neurological and neuropsychiatric manifestations can occur [3,4]. Clinical and experimental data have shown that cortical changes are observed in all forms of liver damage, but it is still unclear which cellular and molecular signaling pathways are affected and at what extend. Since around approximately 30% of normo-ponderal and almost 80% of all overweight people are affected by non-alcoholic fatty liver disease/non-alcoholic steatohepatitis (NAFLD/NASH), this has developed into a silent epidemics in the last few decades [5-7]. With neurological and neuropsychiatric manifestations that vary from acute manifestations (encephalitis, myelitis, encephalomyelitis or Guillain-Barré syndrome [8,9],) to chronic manifestations such as cognitive impairment and dementia [10,11], the exact molecular pathways of this alterations are not completely understood. Theories explaining cortical changes are incriminating the altered permeability of blood-brain barrier (BBB) which allow a large number of small polar molecules to enter the brain, ultimately leading to edema [12-15]. However, insulin resistance, inflammation, hormonal alterations and change in levels of secreted hepatokines [16] have also been reported in this patients.
Some of them will ultimately need a liver transplantation (LT), since grafting is the only definitive treatment for end-stage liver disease, independent of the ethology [17,18]. After transplantation, the BBB permeability impairment can normalize. Regardless, the already acquired brain damage can be permanent. Since 10 to 85% of LT recipients experience central nervous system (CNS) complications (ranging from focal to diffuse brain injury), the main issue with this types of patients is that CNS damage can impair both the short and long-term outcome [19]. In addition, pretreatment with calcineurin inhibitors drugs may exacerbate dysarthria, akinetic mutism, confusion and seizures [20-22].
Given all evidence of cortical changes in patients with liver damage [5,23] and the long-term implication of such changes, a neuroprotective treatment may be considered. Clinical and experimental studies have shown that in other neurological condition like stroke [24], some encephalopathies [25], especially experimental autoimmune encephalomyelitis [26], certain neuroprotective strategies, such as using a Cerebrolysin, a cocktail of low molecular weight peptides and amino acids [27,28]. As the exact mechanisms of its neuroprotective and neurotrophic effects are heterogenous. Reports have attributed its neurotrophic effect to imitations, seemingly through activating the e PI3K/Akt pathway, in a similar manner to o brain derived neurotrophic factor (BDNF) [29-31]. Cerebrolysin's neuroprotective effects have been largely attributed to its role in activating the Sonic Hedgehog signaling pathway [32], but reports have also shown that it is able to lower free radical levels in the cortex and reduce the amount of proapoptotic enzymes [33]. With multiple mechanism involved in the cerebral alterations of liver damage, a single molecular pathway approach for neuroprotection seems not appear to be feasible, but with multiple ways in which Cerebrolysin increases neuroprotection, it seems like a perfect candidate for the task be.”
Line 61: in the sentence: so called “prolonged state of neuroinflammation”, do the authors refer to microglia chronic activation that induces/sustains neuroinflammation? Please review the accuracy of this statement.
The statement was removed from the new introduction
- Material and Methods:
Line 81: authors should be consistent when writing, for instances, “non-viral, non-alcoholic hepatitis” – this is not how it was introduced or discussed along the manuscript.
The content of the methionine/choline chloride food should be added supplementary material.
We added the information in Supplementary table 3
It is not clear when and for how long does the treatment endure. Does it start at week 2? Did it last for 2 weeks, daily ip injections? Any reference study to follow the protocol? Clarify what is the sham, it is unclear what “treatment with serum means” (is it saline?).
The description of the experimental design was improved. It now reads as:
”After relocation to the experimental rooms, mice were given 3 days to acclimatize to the new laboratory conditions. Following the initial 3 days, normal food was replaced with pelleted lacking methionine/choline chloride food (MCD) (MP Biomedicals, Germa-ny) to induce a non-alcoholic steatohepatitis (NASH) [20-22] which we used as a non-alcoholic, non-viral hepatitis model. After two weeks of methio-nine/choline chlorideMCD food intake, mice (n=20) were randomly divided into two groups (n = 10 in each group, with an equal number of males and females per group). All MCD animals received intraperitoneal injections, starting with W2, daily as follows: the control group (MCG) received treatment with serum saline and the treatment group (MCG+Cy) that was given daily intraperitoneal injections with10 mg/kg Cerebrolysin (10mg/kg) (Ever Pharma, Austria). A number of five animals were kept as sham witch were feed normal food and received, intraperitoneal injections of saline starting with week 2 of the experiment. All the mice were fed methionine/choline chlorideMCD ad libitum for 4 weeks, until they were 14 weeks old. During the experimental period a number of 3 animals (sham=1 and MCG=2 died during the anesthesia/ultrasonography procedure).”
Line 150: in this study, authors used BrdU to follow cell proliferation not cell differentiation (cell proliferation is the process of increasing the cell number while cell differentiation is the process of forming a variety of cell types that have specific functions). Please rephrase the statement.
We rephrase the statement. Now it reads like:
“For 3 consecutive days, after the start of the treatment (W2) all animals received two intraperitoneal injections of 5-bromo-20-deoxyuridine (BrdU, 50 mg/kg; Sigma-Aldrich, St. Louis, MO, USA) (8:00 and 20:00).”
- Results:
Line 164: it is missing the number of the week.
We added the number of the week (W2)
Please list the pictures as they are described in the text.
Done
- Please carefully proof-read the manuscript to minimize typos, grammatical errors and coherency of the text:
Line 5: comma after affiliation number
Line 19: end stage vs end-stage
Line 23: Cerebrolisin vs Cerebrolysin
Line 44: his vs these
Line 75: ad libitum vs ad libitum
Line 164: lever vs liver
Line 260: Never the less vs Nevertheless
We hope that all the minor typing mistakes are now corrected
- Main title: I would like to suggest to use “Cerebrolysin” instead of “Neuroprotection”? “Cerebrolysin” in actually the main purpose of this work.
We agree with the reviewer and we made the change.
- US or British English are acceptable, however authors must be consistent throughout the paper.
We hope that all the minor typing mistakes are now corrected
- Make sure that all abbreviations are defined and be consistent throughout the paper.
Done.
- Please check the guidelines regarding “Appendixes and Supplementary Information”, authors must be consistent whether they choose to add sections as Appendix (labelled as Figure A1 or Table A1) or as Supplementary files.
We thank the reviewer for their correction. The materials are supplementary not Appendix. We corrected the oversite.
- Statistical significance: all data that is ∗p < 0.05 is statistical significant. I don’t personally see the purpose of distinguishing ∗∗∗p < 0.001 and ∗∗∗∗p < 0.0001. It is enough ∗p < 0.05 and ∗∗p < 0.01.
We agree that “different” is different (p<0.05) due to the low number of animals (Sham=4, MDG=8 and MCG+Cy=10) we think this is more appropriate as a p =0.011, for example, can increase if the number of animals is increased as is describes in numerous papers on the subject. However if the reviewer insists we will be happy to comply.
- Funding: any grant should be mentioned in this section.
We had no funding for the project, as stated in the original text.
- Acknowledgments: are a place to recognize any contributions made to the paper that do not meet the criteria for authorship.
The reviewer is right. We added dr Smaranda Mitran in this section as she was the one to advice in ultrasonography acquisition and interpretation.